# Causal Strands for Social Bonds—A Case Study on the Credibility of Claims from Impact Reporting

**Jens Teubler * and Sebastian Schuster**

Wuppertal Institute for Climate, Environment, Energy, 42103 Wuppertal, Germany
* Correspondence: jens.teubler@wupperinst.org; Tel.: +49-202-2492-245

**Abstract:** The study investigates if causal claims based on a theory-of-change approach for impact reporting are credible. The authors use their most recent impact report for a Social Bond to show how theory-based logic models can be used to map the sustainability claims of issuers to quantifiable indicators. A single project family (homeownership loans) is then used as a case study to test the underlying hypotheses of the sustainability claims. By applying Bayes Theorem, evidence for and against the claims is weighted to calculate the degree to which the belief in the claims is warranted. The authors found that only one out of three claims describe a probable cause–effect chain for social benefits from the loans. The other two claims either require more primary data to be corroborated or should be re-defined to link the intervention more closely and robustly with the overarching societal goals. However, all previous reported indicators are below the thresholds of the most conservative estimates for fractions of beneficiaries in the paper at hand. We conclude that the combination of a Theory-of-Change with a Bayesian Analysis is an effective way to test the plausibility of sustainability claims and to mitigate biases. Nevertheless, the method is—in the presented form—also too elaborate and time-consuming for impact reporting in the sustainable finance market.

**Keywords:** sustainable finance; social bonds; theory-of-change; Bayesian reasoning; impact reporting

## 1. Introduction

The market for sustainable finance has been growing for the past decade, with an estimated volume of USD 35 trillion of sustainable managed assets in 2020 and a growth of 15% between 2018 and 2020 alone [1]. The role of sustainable finance as an important vehicle for sustainable development is widely acknowledged and translated into policies such as the European green deal investment plan [2]. Sustainable-labeled securities amount to circa USD 250 bn with a large focus on green over social or sustainability bonds [3]. Since the introduction of the first sustainable-labeled securities in 2007, the related frameworks and principles have developed. Many issuers are committed to publish bond frameworks (explaining the rationale and selection of projects). Some also publish external reviews on the eligibility of the assets involved (second-party opinions or SPO) and impact reports on the potential benefits of the assets. Standards such as the ICMAs' green and social bond principles also help to streamline the products in the market. These standards suggest key performance indicators based on a mapping of project categories to the Sustainable Development Goals (SDGs) [4]. They are a tool to assure investors that greenwashing is avoided and to bridge the information gap between them and the issuers [5].

## 1.1. Literature Review

It is unclear whether sustainable-labeled bonds have any direct sustainable impact on the economy and its actors—at least from an investors' perspective [6]. There is reason to believe that these bonds do not necessarily reduce capital costs for sustainable actors in need of capital (the so-called greenium). It is also questioned whether they re-invest sustainable capital into new and additional sustainable projects [7]. Moreover, actual market data suggests that large portions of the increase in green capital to date are not linked to actual growth of a green economy compared to conventional practices [8]. It is therefore likely that the same is true for Social Bonds. On the other hand, there are indications that some investors are willing to forego profits for a greater common value [9].

Moreover, there is also evidence that the environmental performance of green bond issuers is indeed correlated to not only sustainable governance practices (e.g., as shown by [10]) but also to the issuance itself. Flammer (2020) [11], for example, investigated how the issuance of green bonds affected outcomes on a firm level. The author found that, compared to a control group, environmental ratings increase up by 7.3% in the long run, while emissions are reduced by 27.7%. These effects are also larger and more significant for issuers with third-party opinions (certified green bonds).

The impacts of second-party opinions or impact reports by issuers of green, social or sustainability bonds has been, despite these findings, not thoroughly investigated by sustainable finance research. Research here usually focuses on the overall impact of green bonds in the market. Examples of this are studies on the reactions of the stock market to the announcement of green bond issuance [12], investigations of spill-over effects [13] or causal relationships between green bonds and other assets [14]. In a sub-field of this research, many studies investigate whether green bonds are beneficial to issuers or shareholders [15–17] and whether green bonds with some form of external verification (e.g., from a SPO) are higher priced on the market. Dorfleitner et al. (2022) [18], who found this greenium effect, argued that this "[...] indicates that credible and assured non-financial disclosures seems valuable for investors" (Ibid, p. 827). In addition, not only the credibility of the green bond but also the credibility of the issuers itself seem to be an important factor for green premiums. Kapraun et al. (2021) [19], for example, have shown that bonds issued by entities such as governments have a significant advantage here, while bonds issued by corporations require additional green credentials to benefit in a similar manner. However, these credentials come with considerable additional costs for issuers (compared to conventional bonds) and lack standardization and benchmarks [20].

Only few studies investigate the actual environmental or social impacts of sustainable-labeled bonds against practices of reporting. Fatica and Panzica (2021) [21] investigated the relationship between the reporting practices of issuers (e.g., on the use of proceeds) and their environmental impact. The authors actively collected data on the total as well as direct greenhouse gas emissions (GHG). They found that, while emission reductions cannot be directly attributed to projects financed via bonds, issuers show a credible commitment towards climate-friendly behavior. Regarding reporting standards, they also found that issuers with external reviews have an overall lower carbon intensity (signaling a higher commitment to climate-friendly practices). Tolliver et al. (2019) [22] looked at the overall green bond market growth between 2008 and 2017 and compared it to the fraction of issuers that report on the environmental impacts. They found "[...] that there here has been a massive gap between green bonds market growth (i.e., in issuances and outstanding) and reported proceeds allocations" (Ibid, p. 7). According to this analysis, international financing institutions (IFIs) "[...] provided the bulk of environmental impactful green bond finance" (ibid, p. 12). The authors attributed this to the greater financing schemes of these institutions as well as their commitment to joint harmonized frameworks.

The current discussion on appropriate methodologies for impact assessments and reporting focuses on the savings of greenhouse gas emissions (GHG) for climate change mitigation and is driven by practitioners (e.g., [23,24]). Practices here lack consistency

regarding the methods, data and indicators used [5]. A notable exception is a recent scientific paper by Gibon et al. (2020) [25]. The authors investigated how life-cycle assessment (LCA) methods and databases can be applied for a more robust environmental impact assessment from green bond projects. Using the example of renewable power plants, the authors estimated life-cycle-wide avoided GHG emissions for different baselines and compared the results to the use-phase estimates reported by the issuers. They found that the selection of the reference case has a strong influence on the results with issuer estimates of avoided emissions being 11% higher than LCA results for an average electricity mix. Within the investigated LCA-based approaches, this baseline was in turn ca. 75% higher than a baseline using a consequential LCA approach with marginal changes to the electricity mix. Given this fact as well as the fact that results vary highly within technologies, the authors are convinced that investors currently do not have sufficient information to select the most effective bonds.

Since Social Bonds are new to the market, there is currently no common methodology to provide evidence for their efficacy. It is well understood though, that evaluating the outcomes of Social Bond projects is more challenging compared to Green Bonds, especially regarding the human rights perspective. Park (2018) [26], for example, argues that the available indicators obstruct accountability because they are inconsistent, incompatible and incomplete. More importantly, most human-rights-related impacts occur in the long term, while the considerations of financial actors are driven by a short-term perspective.

Nonetheless, the underlying rationale of Social Bonds for social change through financing programs and projects is very similar to the more established practices of impact investing, impact bonds or blended finance operations. Although Social Bonds do not address the direct financing of stakeholders, many of the findings from the literature on blended finance are applicable. The most recent and most comprehensive body of research on this subject can be found in OECD Development Co-operation working papers, such as Boiardi (2020) [27] on impact management and measurement, Andersen et al. (2021) [28] on additionality of blended -finance operations and Habbel et al. (2021) [29] on the use of evaluation methods such as theories-of-change (ToC).

Theories-of-change (ToC) have frequently been used for evaluation purposes in the field of impact investing. Jackson (2013) [30], for example, criticized early on that "[…] current practices in the evaluation of impact investing still tends to focus on counting inputs and outputs, and telling stories" (Ibid, p. 99). He argued that the use of development evaluation practices (and theories-of-change in particular) provides the industry with a more comprehensive frame for understanding the function of evaluation. It also comes with relevant data collection and analysis methods and helps to prioritize the actual positive changes for the beneficiaries. Although ToC approaches certainly fulfil this purpose, they are not sufficient measurement tools themselves, as Verrinder et al. (2018) [31] showed. The authors investigated three cases in which ToCs were used. They found that they indeed generate an understanding of the mechanisms involved and create a plausible explanation of how stakeholders contribute to desired outcomes. The process itself "[...] results in a stronger belief, internally and externally, in what the organizations are doing" (Ibid, p. 8) and helps to identify what ought to be measured. Measuring these indicators is considered to be an additional challenge.

### *1.2. Research Question*

From a societal perspective, it is crucial to understand if issuers contribute to overarching sustainability goals and if these contributions can be attributed to capital by investors. External reviews and assessments play an important role here, as they corroborate, test and sometimes quantify the claims of the issuers. The quantification of these desired outcomes is still in its infancy—at least in impact reports for Social and Sustainability Bonds. Although this is true for all dimensions of sustainability, the identification and quantification of potential social benefits constitutes the largest challenge yet.

Social Bonds are often associated with institutional issuers (or socially committed stakeholders such as NGOs) that have experience in designing social programs. This can be an advantage compared to mainstream actors in the financial market, as societal goals and target groups have already been identified. However, any issuer claiming to achieve societal goals also needs to show how progress was (ex-post) or can be made (ex-ante). This causal link between intervention and desired outcome is the focus of the paper at hand. Without it, no contribution by the issuer can be claimed confidently (nor is it reasonable to quantify indicators describing this effect). Consequently, there would also be no attribution by investors in the secondary market (regardless of the question of actual additionality).

We discuss this issue with the help of the case study of the NRW.BANK Social Bonds [32]. These four bonds have been issued 2021, refer to a three-year portfolio (bond-to-pool) and re-finance the NRW.BANK in four areas of intervention. NRW.BANK is a promotional bank that supports the German federal State of North-Rhine Westphalia in the completion of its structural and economic policy tasks. Most of its customers are house banks and other promotional intermediaries such as the German KfW, but direct lending occurs as well.

The associated impact report #2 by Teubler (2022) [33] and its underlying methodology [34] were published in 2022. We present how a Theory-of-Change (ToC) approach was employed here to develop a plausible and probable causal connection between the inputs of the bank and the associated Sustainable Development Goals (as defined by the issuer in its framework). As the overarching goals were pre-defined by the issuer, the analysts (here, the corresponding author) show how inputs and goals can be connected before estimating what type of effect occurs. This developed linear model was then used to identify, present and quantify indicators along the assumed cause–effect chain.

This study is divided into two parts and focuses on the follow-up process of validating the claims of the ToC. First, we look more deeply into one of the assessed schemes: homeowner ship loans to purchase or build new homes for low- and mid-income households. We investigate how both the claims by the issuer as well as the (subsequent) claims of the ToC framework hold up under scrutiny. To this end, the ToC for these loans is translated into three separate hypotheses that are tested with the help of a Bayesian Analysis (BA). We look for evidence for and against the main hypotheses of the ToC cause–effect relationship.

Secondly, we discuss how the results of the BA can inform both the issuer and the analyst in terms of future data needs and improvement of the stated causal claims. Using the current degree of belief in the validity of each hypothesis, we identify what changes to the causal assumptions and what type of issuer data would be required for that.

These two parts correspond to three research questions (see Table 1), which aim to further develop the methods used and to discuss if and how the methods can be generalized.

**Table 1.** Research questions of this study.

| No. | Research Question |
|---|---|
| 1 | To what extent does the belief in the causal claims of issuer and analyst (ToC claims) warrant that homeownership loans ensure access to affordable housing and reduces poverty? |
| 2a | Given our current degree of belief (1), what primary data would improve the credibility of the ToC claims? |
| 2b | Given our current degree of belief (1), what changes to the causal assumptions would result in a higher credibility of the ToC claims? |

## 2. Materials and Methods

### 2.1. Materials and Methods for ToC

There are numerous methods and variations that can be associated with theories-of-change. These types can be broadly differentiated into scholarly methods concerned with social program design and evaluation (e.g., program theory as suggested by Weiss and Rogers [35,36]) and practitioner approaches concerned with changes by or within organizations (e.g., described by [30,37,38]). Unsurprisingly, not only are these methods applied in different applications and contexts, but the terminology differs as well.

The ToC for the NRW.BANK Social Bond impact report relies on a complicated linear logic model as defined by Rogers (2008) [39]. The model is linear because it is one-directional. Outcomes are rooted in interventions but cannot affect interventions, nor is any other form of circularity involved. The model is also complicated (in opposition to complex or simple models) because multiple causal strands can affect multiple actors. The characteristic of linearity is crucial for applying the methods described in this paper, but they also limit their real-life applicability.

The further characteristics of the ToC discussed here are its use of an implicit stakeholder theory and the omittance of pre-conditions for change (see [40] for an overview of the core principles of theory-driven evaluations). The first attribute is an outcome of the impact assessment process itself. The analyst is required to be an external reviewer for an ex-post evaluation. Thus, he is not involved in decisions on the core logic of the Social Bond Framework. The issuer and not the analyst defines what overarching sustainability goals the projects aim to contribute to (corroborated by a third party). The second attribute (omittance of pre-conditions) is an intentional simplification of the developed ToC model. It allows to formulate a lean and minimalistic cause–effect relationship that can be quantified and tested more easily. The influence of surrounding systems is acknowledged, but not explicitly accounted for. Instead, a so-called responsibility ceiling is visualized that constitutes the area of direct influence by the lender (what activities and outputs must materialize for the desired outcomes). However, both simplifications restrict the likelihood of truth claims, as pattern-seeking bias (searching for causal connections to the implicit stakeholder theories) and cognitive bias (shortcuts from omitting important pre-conditions) increase the probability of erroneous conclusions.

#### 2.1.1. ToC Framework

Six entities in the outcome pathway of the ToC framework are distinguished: inputs, activities, outputs, intermediate outcomes, long-term outcomes and impacts. Figure 1 shows how these components are understood in the ToC and describe how they are used in the context of one of the causal strands discussed in this paper. Appendix A.1 summarizes how the ToC was developed and the impacts investigated.

This paper focuses on the overall narrative as it relates to the cause–effect chain between the interventions (inputs) and their contribution to the Sustainable Development Goals (impacts). The hypothesis (h) to be tested describes how the desired outcomes on the societal level can, or even ought to, be achieved. Because logical fallacies and biases could also be hidden in the assessment part of the process, the quantified indicators are not part of the investigation. However, any background data used to quantify these effects can be considered evidence for or against the consequential probability of h being true.

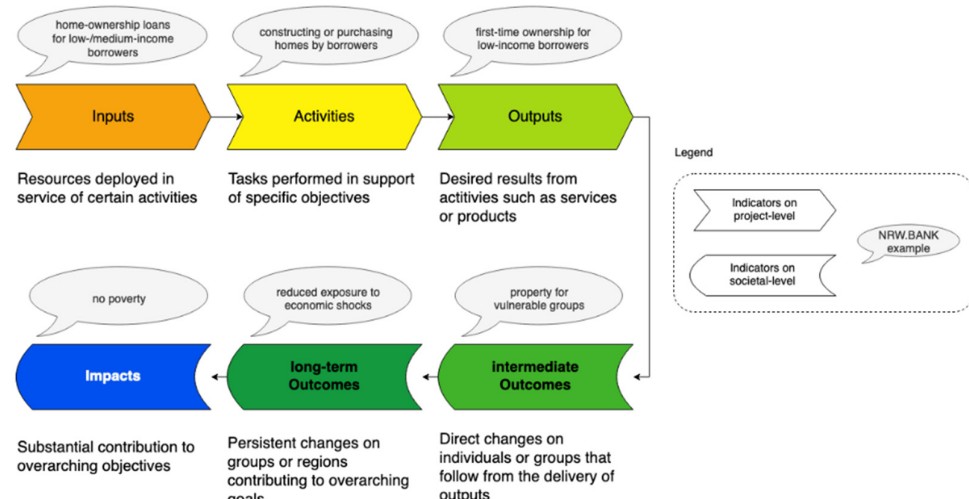

**Figure 1.** Entities in the ToC for NRW.BANK Social Bonds (in: "Impact Assessment Methodology for NRW.BANK Social Bonds" [34] and adapted from [40–42]).

### 2.1.2. Selection of the Case Study

The NRW.BANK loans are associated with four areas of intervention: Affordable Home Ownership, Access to Essential Services: Education, Access to Public Goods and Services and SME Financing and Employment generation.

We selected the first intervention (Affordable Home Ownership) because this program is restricted to one type of loan. It is also associated with the most risks and hazards for achieving the desired outcomes and refers to policies that have been investigated in the literature before.

### 2.1.3. Primary (Issuer) Data for the ToC

The loans designated for affordable homeownership of families and singles are called NRW.BANK.Wohneigentum. They were designed after the federal program Baukindergeld, which is the main program of comparison for the study at hand. However, NRW.BANK.Wohneigentum does not include additional grants to borrowers and is not restricted to families.

The dataset covers the years 2018, 2019 and 2020 and includes circa 3,400 loans with a total volume of EUR 396m. Each loan is associated with its volume, the postal code of the borrower and the postal code of the property to be acquired. The issuer also provided the analyst with an estimation on the share of borrowers with an annual gross income of less than TEUR 50 (entire household) compared to borrowers with incomes of up to TEUR 75 and more (the number of children increase the upper limit up to which a loan application is accepted). The analyst then used the data on postal codes to identify the population density at the location where the property is acquired.

### 2.2. Materials and Methods for Bayesian Analysis

Bayesian approaches, such as Bayesian Networks, Bayesian Confidence Intervals and Bayesian Analysis, are methods of statistical inference. They are popular in a wide range of scholarly or scientific areas, such as computer science [43], medicine [44], biology [45], history [46] and psychology [47].

Bayesian Analysis or Bayesian Reasoning describes the posterior probability of a hypothesis being true. It is conditioned on our prior background knowledge as well as the consequent likelihood of evidence, given that the hypothesis is true or false. Its long mathematical formula for propositions defines the posterior probability as:

$$P(h_0|e.b) = \frac{P(h_0|b) \; x \; P(e|h_0.b)}{\sum_n P(h_n|b) \; P(e|h_n.b)} \qquad (1)$$

where

P: Probability;

$h_0$: Hypothesis to be tested;

$h_n$: Alternative hypotheses;

b: Current background knowledge;

e: Evidence.

When testing one specific claim against all other claims, this formula can be simplified by summarizing the denominator in (1) to account for the hypothesis *h* being false ($\neg h$):

$$(h|e.b) = \frac{P(h|b) \; x \; P(e|h.b)}{[P(h|b) \; x \; P(e|h.b)] \; + \; [P(\neg h|b) \; x \; P(e|\neg h.b]} \qquad (2)$$

This relationship can also be expressed in the form of odds, as shown in (3):

$$\frac{P(h|e.b)}{P(\neg h|e.b)} = \frac{P(h|b)}{P(\neg h|b)} \; x \; \frac{P(e|h.b)}{P(e|\neg h.b)} \qquad (3)$$

Appendix A.2 further describes the components of a BA and discusses how they are understood in the context of this study.

### 2.2.1. Reasons for Conducting a Bayesian Analysis

We used the Bayes Theorem, or rather conducted a Bayesian Analysis (BA), to test the ToC model for the following reasons:

1.  ToC is a heuristic method for explaining stakeholder theories on causes and effects. It is seldom based on robust empirical studies investigating the claims. Although BA can be used with empirical data only, it is also applicable to epistemic justification or a mixture of both.
2.  BA requires the researcher to formalize and explicitly state all arguments for causation, even if they are (originally) based on intuition or biases. This common ground allows for a more informative and constructive discussion with other scholars.
3.  BA provides information on the likelihood of a hypothesis being false. This allows the user to compare competing hypotheses, and in the case of a ToC model, to qualify which causal strands are more robustly attested to.
4.  In BA, good evidence can overcome bad odds. This suits its application for a ToC model because ToC usually deal with interventions that are explicitly introduced to cause changes in the existing system.
5.  The simple mathematical connotations of a BA can be translated into plain English, and plain English can be used to estimate probabilities. Therefore, statistical reasoning informs the conclusions, but its semantic depiction also allows to identify logical fallacies easier than statistical methods of observation.

### 2.2.2. The Result of BA in the Context of the Study

Bayesian Reasoning deals with the question of how, why and to which degree our beliefs change given the evidence. Its original formula refers to the probability of an event *A* occurring given that a condition *B* is true and compared to the odds of *B* being present if event *A* occurs:

$$P(A|B) = \frac{P(B|A) \; P(A)}{P(B)} \qquad (4)$$

Or odds form:

$$O(A_1 : A_2 \,|B) = O(A_1 : A_2) \times \frac{P(B|A_1)}{P(B|A_2)} \tag{5}$$

where $P(A)$ and $P(B)$ are the probabilities of observing each event without any given conditions (and B having a non-zero probability of occurring). This entails that, for Bayesian Reasoning, causes are not necessarily pre-conditions for subsequent effects (or can at least looked at from both perspectives). In fact, Bayes Rule even works when evidence is considered that refers to a point in time before the event occurred (e.g., in the case of having evidence for a murder suspect preparing the scene before committing the deed). This original version of BT is still used for a variety of purposes (e.g., odds of a correct diagnosis). In other areas such as archeology, it is more common to apply the epistemological interpretation of BT, as shown by Formula (1). This probability describes, and somewhat quantifies, the degrees of belief with and without the given evidence.

We preferred the latter connotation for our study, as it fits more closely with our understanding of what the result of a BA can be for a heuristic map of causal strands in a ToC. We based our definition of the results of BA in the study at hand on Carrier (2012) for BA ascribing to "warranted beliefs" rather than "truth" [46]:

> *The results of a Bayesian Analysis on the credibility of claims in a linear Theory-of-Change model describe, given our current knowledge, the degree to which our belief is warranted, that the desired outcomes are achieved because of its stated interventions.*

We also used Carrier's rule of the greater knowledge (rule A) for defining the most appropriate reference class (see Appendix A.2) and rule from the stronger argument (rule B or a fortiori) for estimating the probabilities in general. These two rules ensure that the most relevant reference system is chosen before estimating the prior (rule A) and that our results describe a conservative case for the claims to be true (rule B).

### 2.2.3. Semantic Depiction of BA for the Study

The depiction of a BA in plain English is helpful when the researcher tries to avoid an erroneous operationalization of the method. Expressing ranges of certainties and describing the data in Bayesian terms makes it easier to spot potential conjunctions where there should not be ones (especially between priors and consequents). It also helps in identifying logical fallacies. A semantic depiction of certainties or degrees of belief is what we already do anyway when applying our knowledge and our intuition to draw causal inferences from propositions (e.g., by expert guess). We found it, therefore, helpful to translate the epistemic interpretation of BA (formula (2)) into plain English, as shown here and adapted from Carrier (2012) [46] (see Figure 2):

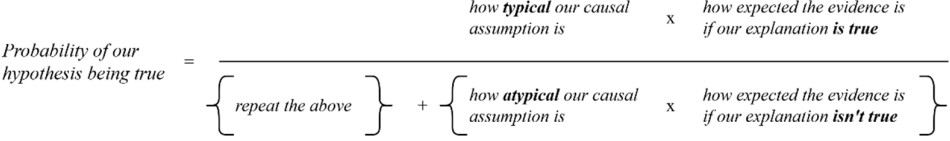

**Figure 2.** Epistemic interpretation of Bayesian Analysis in the context of the study (own development based on [46]).

We also adapted Carrier's "canon of probabilities" to our purpose by applying its range for semantic description of probabilities and assigning it to the number of people in a sample for which its hypothesis is expected to be true (see Table 2). If, for example, we find it very improbable that a borrower is better off economically with the loan than without it, only 5 in 100 borrowers are expected to benefit in this manner (odds of 1:19).

**Table 2.** Semantic scale for probabilities and distributions in the study (our own compilation based on the "canon of probabilities" of [46]).

| *Qualitative scale for probability* (P) | Application for study (odds) |
|---|---|
| virtually impossible (0.0001%~0%) | false |
| extremely improbable (1%) | true for 1 in 100 people (1:99) |
| very improbable (5%) | true for 5 in 100 people (1:19) |
| improbable (20%) | true for 20 in 100 people (1:4) |
| slightly improbable (40%) | true for 40 in 100 people (2:3) |
| even odds (50%) | true for 50 in 100 people (1:1) |
| slightly probable (60%) | true for 60 in 100 people (3:2) |
| probable (80%) | true for 80 in 100 people (4:1) |
| very probable (95%) | true for 95 in 100 people (19:1) |
| extremely probable (99%) | true for 99 in 100 people (99:1) |
| virtually certain (99.9999%~100%) | true |

## 3. Results

### *3.1. Causal Strands of ToC*

The ToC for home-ownership loans is based on the implicit stakeholder theory of the issuer. NRW.BANK published a Social Bond Framework before issuance, which provides the following information and makes the following assertions [32]:

1.  The target population are economically disadvantaged persons (as defined by the issuer) or families with (a) a taxable household income below TEUR 75 p.a. plus TEUR 15 per child under 18 or (b) a taxable household income below TEUR 60 p.a. for couples, TEUR 30 p.a. for singles as well as TEUR 12 p.a. per child under 18.
2.  The loans are asserted to be aligned with SDG 1 (no poverty) and SDG 11 (sustainable cities and communities).
3.  Loan applicants for constructing or purchasing owner-occupied residential properties can submit one funding application per property and household (including costs for modernization, land and outdoor facilities) for up to 50% of the total costs (loans from other programs are explicitly encouraged).

The corresponding SPO [48] reviewed this information and corroborated the claim of its contribution to sustainability goals. They consider home-ownership loans to be a significant contribution to SDG 1 for reducing poverty.

### 3.1.1. The ToC for Home-Ownership Loans

The following Figure 3 shows the resulting ToC from the current impact methodology. Its associated narrative reads:

*"NRW.BANKs' homeownership loans contribute to better access to affordable housing (SDG 11) and reducing exposure to economic shocks (SDG 1) for low- and medium-income families by improving the affordability of living space, increasing disposable income for households and stabilizing rents. To achieve these goals, the financing directly enables first-time homeownership for vulnerable groups and a decrease in living expenses for the borrowers. In many cases, the borrowers increase the supply of rental living space or rent-out living space after acquisition at rates that are at or below the current renting index. In terms of hazards, it cannot be ruled out that living expenses might instead increase for some burrowers (if for example the trade-off between former rental rate and loan costs is unfavorable). There is also a risk that the freed-up living space is rented at considerable higher rates than before"* [34].

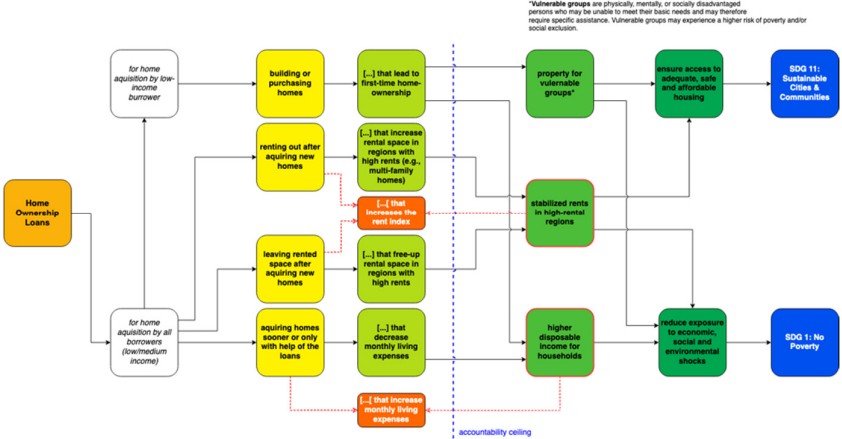

**Figure 3.** Theory-of-Change for NRW.BANK home-ownership loans [34].

There are two causal strands originating from two different actors: (a) the larger group of all low- or medium-income borrowers and (b) the sub-group of only low-income borrowers (which are considered to be vulnerable groups). For impact SDG 11, only low-income borrowers build or purchase homes that lead to first-time ownership and subsequently to property that constitutes access to affordable housing (with the implicit assumption that mid-income borrowers have that opportunity anyway). Mid-income actors only contribute to this goal if there is sufficient evidence to believe that their activities have a positive effect on the renting index. SDG 1 is divided into two sub-strands as well. Again, low-income borrowers acquire property that, in turn, reduces their exposure to economic shocks. However, for mid-income borrowers, these effects are only asserted if their disposable income is increased as a result of no longer paying rents for their previous homes. The second strand (which can include low-income borrowers) is, therefore, more in line with preventing poverty rather than reducing it.

### 3.1.2. Hypotheses of ToC Case-Study (Truth Claims)

The next step in this analysis is to define hypotheses that can be tested with the help of BT. As the ToC here is strongly intertwined (connections between both sustainability goals and target groups), this process needs to be framed by clear guidelines. Our goal is to construct hypotheses that have the least assumptions, as all additional (ad hoc) assumptions would have to be tested as well.

First, we looked at those parts of the ToC that can be asserted to be true with almost 100% certainty (prior probability of more than 99% and low chance of finding contradicting evidence), as these causal statements do not need to be tested. We assumed this to be the case on the right side of the ToC, where long-term outcomes contribute to impacts. These outcomes are direct paraphrases of sub-targets in the SDG framework and, therefore, attested to be true by reliable sources. Reducing exposure to economic shocks is one important requirement to reduce poverty [49], and access to affordable (and adequate) housing is needed for inclusive cities and human settlements [50].

Likewise, we assumed that the loan program of the issuer does what it is supposed to do. That is, the bank provides loans to certain income groups that are ear-marked for acquiring residential homes. We do not know if these acquisitions develop how the ToC assumes they do. However, we can assert with almost 100% certainty that the money loaned is used for that purpose, reaches the defined groups and there are obligations to provide evidence.

The third causal connection we want to exclude from the BA is the assumption that stabilizing or even reducing rents contributes to both affordable housing and reducing exposure to economic shocks. This can only be considered as almost certainly true if we

consider the context. All loans are paid out to residents in the State of NRW with a very high likelihood that they are used for homes in that state (or at least in Germany). Notwithstanding the question on how those homes affect the renting index (which is part of Hypothesis 3), it can, in our opinion, safely be assumed that almost any region in Germany, and NRW, has a history of increased rents and housing shortages (especially since 2014, according to [51]). We conclude that there is a very high probability that stabilizing rents leads to benefits for the target groups. These benefits are (among other possibilities) better access to affordable housing and reducing exposure to economic shocks.

The fourth and final exclusion refers to the relationship between first-time homeownership for vulnerable groups (if true) and an increase in their share in the population. Although there is a time gap between these two conditions, we cannot conceive a world where this is not almost certainly the case.

There are also ad hoc assumptions that are necessary to conduct the assessment. The narrative of the ToC does not specify at which point in time intermediate outcomes are achieved. This is crucial for all three hypotheses, but it is, in particular, relevant for the causal link between property acquisition, monthly expenditures and disposable income (Hypothesis 2). Any property under the loan scheme requires debt capital to be paid back over several years, while most borrowers probably also saved money beforehand to afford the equity share of the acquisition costs. This means that the initial change in disposable income at the beginning can be both positive (no more additional savings) and negative (double payments for rents and loans at the same time). After this period, monthly expenditures are constant (repayment period with interest) for a long time before dropping to almost zero at the end (property taxes and maintenance costs). We assumed that the period of almost constant monthly credit costs is the most relevant for all three hypotheses. We also assumed that this time frame starts no later than one year after the borrowers moved into their new home and ends before the full loan has been paid back.

The second ad hoc assumption relates to hypothesis 1 only, in which property for *vulnerable groups* is acquired and this term needs to be further defined. Vulnerability can be considered a dimension of poverty, because poor households are usually "more exposed to risks and less protected from it" [52]. For the purpose of the assessment, vulnerable groups are slightly above or below the risk-at-poverty threshold in Germany. However, an increased exposure to risks can also lead to poverty, if households, in an attempt to avoid that exposure, take costly preventive measures (ibid). This is particular important if certain household types are more likely to avoid high risk, but high return activities, such as acquiring a home with the help of a loan (ibid). We defined that the following characteristics can be applied to this second effect: elderly, single parents, disabled or invalid persons and migrants.

Given these certainties (we assumed were held true regardless of the ToC) and these ad hoc assumptions, the following three hypotheses are tested with help of the Bayes Theorem:

**h₁**: *Home acquisition by low-income borrowers leads to first-time homeownership for vulnerable groups.*
**h₂**: *Home acquisition by low- and medium-income borrowers increases the disposable income of former tenants in the medium term, which reduces their exposure to economic shocks.*
**h₃**: *Home acquisition by medium-income borrowers increases and frees-up rental space in regions with high rents, which stabilizes the rental index in these regions.*

These general causal statements are tested against our background knowledge (constituting the prior probability) and the likelihood that the evidence found strengthens or weakens the argument (constituting the consequential probability). Note that these statements refer to the entire sample (low- and mid-income borrowers according to the issuer) or at least to a well-defined fraction (low-income borrowers according to the issuer). Any posterior probability above 0% and below 100% could therefore be translated into an estimate on how many borrowers benefit as intended compared to the rest. For example, a

posterior probability of 42% for $h_1$ translates into our expectation that 42 out 100 loans lead to the desired outcomes. As a rule of thumb, and consistent with other Bayesian studies, we regarded each hypothesis as true, if most loans (more than 50%) compared *to some* loans (<50%) work as predicted by the ToC. However, we also discuss the implications of posteriors at the lower or higher end (e.g., by which means the bank could increase a low posterior).

### 3.1.3. Indicators of ToC Case Study

Although the indicator assessment is not a necessary part of testing the hypotheses, they can inform on the probability of them being true (either as part of the background knowledge or as evidence). Table 3 lists all quantified indicators from the impact assessment (for a full list of all indicators in all assessed areas, see [33]).

**Table 3.** Quantified indicators for the home-ownership loan program in NRW.BANK Social Bond #2021 (from Teubler (2022a) [33]).

| Indicator | Value | Causal Link | Model |
|---|---|---|---|
| property for vulnerable groups | 1471 housing units | int. outcome on $h_1$ | share of households with low incomes |
| borrowers with decreased living expenses in cities | 819 borrowers | output on $h_2$ | property in regions with high population density in NRW |
| homeownership in regions with high rents | 110 housing units | output on $h_3$ | property in 18 regions in NRW where laws are in place to limit the increase in rents |
| homes built | 1403 housing units | activity | share and costs of housing units based on NRW statistics |
| homes purchased | 1756 housing units | activity | share and costs of housing units based on NRW statistics |

None of the indicators shown here ensures the validity of the claims, as it was not possible to measure any direct effects regarding the three hypotheses. Additionally, only three out of six indicators show any connections to the causal conditions to be tested. However, they all already point to only a fraction of the entire sample (e.g., only 110 out of 3359 loans are expected to lead to a rent stabilizing effect).

These indicators can therefore also be understood in Bayesian terms. If these indicators were all what we know and assumed to be true, we would expect that 100 out of 100 of low-income borrowers benefit on $h_1$, 26 out of 100 of all borrowers benefit on $h_2$ and 13 out of 100 of borrowers in cities benefit on $h_3$.

Although the indicators themselves cannot be used directly for our analysis, the underlying primary and secondary data can be relevant as background knowledge.

### 3.2. Background Knowledge and Evidence

The background knowledge of the study is summarized (and referenced to) in Table A1 in Appendix A.3. It contains all information and data provided by the issuer or found in the literature. It also indicates if and where this information is then used later to estimate priors and consequents. These elements are abbreviated and numbered as $I_n$ (for information). Their use is abbreviated as $h_n.b$ for background knowledge on the priors ($P(h|b)$) for hypothesis $h_1$, $h_2$, $h_3$ and as $h_n.e$ for the expected evidence on the consequents ($P(e|h.b)$ and $P(e|\neg h.b)$).

The most relevant literature sources are evaluations of the federal loan program Baukindergeld [53,54] and its comparison with a sample of families moving into their own property [55], the social report for the State of NRW [51], real estate reports by the issuer [32,56], real estate studies on behalf of Accentro Real Estate AG [57–59] and further statistical data on living costs and income [60–62].

### 3.3. Warranted Belief in Hypothesis 1

**h₁:** *Home acquisition by low-income borrowers leads to first-time homeownership for vulnerable groups.*

### 3.3.1. Prior for h₁

This section defines the prior probability of h₁ being true based on the available background knowledge. The background knowledge for hypothesis 1 is ambiguous. On the one hand, given that residential property is acquired, this property is very probable to be first-time homeownership for low-income groups (I_34) and thus increasing the homeownership ratio in NRW for this target group.

Most loans require equity and all loans the means to pay the monthly rates—at least, if we assume that the lender reviewed the financial resources of the borrowers (which is a pre-requisite for providing the loans). It is therefore expected that a considerate portion of the low-income borrowers are at the higher end or above the at-risk-of-poverty threshold. By opposition, it is also plausible to assume that some of these borrowers have characteristics that make it more likely that they belong to a vulnerable group regardless of that fact (such as handicapped people, people in precarious jobs and single people with children).

For now, we can only consider the income and rental situation of first-time owners for our prior of h₁. We know from I_1 to I_3 that all non-singles in the sample of low-income borrowers are below the at-risk-of-poverty threshold for Germany and that 87% of first-time owners belong to that group (I_4). We also know that in a comparable sample of Baukinder households, at least 23% are below or very near to a risk-at-poverty threshold (I_28).

Regarding first-time ownership, we have reason to assume that this rate is very high. While the portion of first-time owners that move out of rental space on average is already high (71% according to I_5), almost all Baukindergeld households lived in rental situations before acquiring their property (98% according to I_12). We argue that it is very unlikely that these low-income borrowers had property before or lived in a rental situation, while owning homes and that we can therefore assume that 98% of low-income borrowers are first-timer owners of property.

Combining these probabilities leaves us with a prior of 98% out of 87% of the low-income sample and therefore with P(h₁|b) of 85% (85 out of 100 low-income borrowers are generally expected to belong to a vulnerable group acquiring first-time property).

### 3.3.2. Consequent for h₁ and ¬h₁

We focused our investigation of the consequents on the predictors of vulnerability in the group of borrowers and concluded it with information on available equity for low-income households. Our categories for the consequents on h₁ and ¬h₁, respectively, derive from the available evidence in the background knowledge and a definition what type of borrowers belong to a vulnerable group (see second ad hoc assumption in Section 3.1.2.). The following predictors for vulnerability are each compared to the overall population of first-time homeowners: single-income earner or families with many children (CHI), migrants or persons from second-generation migrant families (MIG) and disabled or handicapped persons (DIS). There are also borrowers that might belong to more than one group. However, we sought to find the minimal, or most conservative, degree of belief in the hypothesis and such groups would amplify effects already attested to.

The only relevant evidence we found (if at all) was drawn from the evaluation of the federal Baukindergeld scheme [53]. This is also why only characteristics were selected that would be expected for families with children in their homes (e.g., elderly might be part of these households but are not expected to apply for the program themselves). To accommodate for the fact that relevant information is not available, we also developed consequents for an argument from silence (SIL).

Regarding parenthood, there is no information in the Baukindergeld evaluation on the actual share of single parents or single-income earners (CHI). The only information we found describes the share of Baukindergeld households with 1, 2, 3 or 4 and more children. As Baukinder households do not differ in this regard from the general population (I_26), this evidence is considered slightly less likely on $h_1$ as it is on $\neg h_1$ (odds of 90:100).

We also have no direct information on the share of migrants in the sample (MIG). However, Baukindergeld households are more likely to have German citizenship than first-time owners with families in general (I_26). This is fully expected on $\neg h_1$ (100%), but slightly less probable on $h_1$. Given the knowledge that non-German borrowers also have at least medium-level incomes and the small difference between the samples (12% compared to 15%), we think that the odds of 80:100 are justified here a fortiori (which is what we would expect from 12:15).

As there is no evidence for disabled or handicapped persons in the Baukindergeld sample, there is also no evidence to be included in the consequent (odds of 100:100). However, we found it at least somewhat conspicuous that an evaluation study for the Baukindergeld households collecting socio-demographic data of borrowers would not mention vulnerable groups such as single mothers or handicapped persons. It is of course possible that these characteristics were not evaluated at all but find it more likely that these characteristics were not prevalent enough in the sample to even mention them. Estimating the odds for this is difficult, so we assumed an upper and lower bond, using then the lower bond as a fortiori estimate. Given that the evaluation questions focused on the support for homeownership by families but also on the housing market effects, it can be argued that these characteristics were not very relevant for the evaluation and therefore expected on both $h_1$ and $\neg h_1$. We assumed that this upper bond is at 90% on $h_1$ compared to 100% on $\neg h_1$. On the other hand, data on the income and number of family members are mentioned in other contexts, which is why a more conservative estimate on $h_1$ would be 60%. This leaves us with odds for the argument of silence of 60:100.

The final piece of evidence on $h_1$ refers to the equity of the borrowers (EQU). We expected on $h_1$ that borrowers from vulnerable groups would have more problems in providing the starting capital for property compared to the overall population of first-time owners. There is no information on the actual available capital of households in both samples. We only know that a higher portion of Baukindergeld borrowers needed to save capital before acquisition (61% compared to 37%), but that the duration of saving does not differ between both groups (I_34). This evidence only relates to the topic in a tangential manner and would only be slightly in favor of $h_1$. We found it therefore reasonable to argue that the likelihood for this piece of evidence is equal on both propositions (100:100).

Overall, the following odds are multiplied to calculate both *consequent*s:

$$\frac{P(e|h_1.b)}{P(e|\neg h_1.b)} = \frac{9}{10} \; (CHI) \times \frac{8}{10} \; (MIG) \times \frac{1}{1} \; (DIS) \times \frac{6}{10} \; (SIL) \times \frac{1}{1} \; (EQU) = \frac{43.2}{100}$$

The resulting consequents are 43% for $P(e|h_2.b)$ and 100% for $P(e|\neg h_2.b)$.

### 3.3.3. Posterior for $h_1$

Based on the estimates for both priors and consequents, the following degree of warranted belief in $h_1$ was established:

$$P(h_1|e.b) = \frac{85\% \times 43\%}{[(85\% \times 43\%) + (100\% \times 15\%)]} = 71\%$$

The posterior probability for $h_1$ being true is therefore assumed to be at circa 70%, which is clearly above the credibility threshold. Out of 100 borrowers in the smaller *low-income* sample, 71 borrowers are expected to belong to a vulnerable group acquiring first-time property. By comparison with the entire sample, no more than 33% are expected to benefit from the program in this manner. Note that this value does not relate to the credibility of the claim itself as hypothesis 1 already only considers households with a gross income of less than 50 TEUR.

### 3.4. Warranted Belief in Hypothesis 2

**H2:** *Home acquisition by low- and medium-income borrowers increases the disposable income of former tenants, which reduces their exposure to economic shocks.*

#### 3.4.1. Prior for $h_2$

The background knowledge for hypothesis 2 is insufficient to come to a clear conclusion for the prior probability. Homeowners certainly benefit in the long run (I_8) when they no longer pay rents (98% of borrowers are former tenants according to I_12) and the loan has been repaid. For the medium term, they are clearly at a lower risk of paying excess housing costs (I_6) and have, on average in Germany, lower monthly housing costs compared to tenants on average (I_7) (The interest rates were very low at the time of the study, and even lower for loan schemes such as the one investigated in this paper. These rates are currently increasing due to high inflation and other factors, while housing prices are dropping. These claims might therefore be less robust when looking at the housing market in 2022 compared to 2018–2020). This indicates that the prior for hypothesis 2 is very high. However, for the hypothesis to be true, a sufficient share of an actual group of borrowers (or a group that can be compared to them) need to benefit from the program by improving their disposable income.

Unfortunately, there are no data that directly compare the cold rents of former tenants to their repayment and interest expenditures after home acquisition in NRW (which would be the desired reference class). Instead, the observed cold rents in Germany (I_9) were compared to the annuity costs of homeowner in NRW (I_10). Both represent a fixed portion of income that needs to be spent monthly, but do not include all costs and are based on different sources and calculation methodologies. They cannot be used to calculate a prior directly but might indicate to which end of the spectrum the prior leans to for a conservative estimate.

We started by assuming that the chances are rather high for the hypothesis being true based on the average statistics and then look at the differences between annuity and cold rents for an update of that assumption. At the lower end of the spectrum, cold rents and annuity (as in credit costs for repayment and interest) are at EUR 334 per month (p.m.) and EUR 474 p.m., respectively (I_9, I_10). At the higher end of the spectrum, cold rents are at EUR 1084 p.m. and credit costs at EUR 1286 p.m. Looking at the average, cold rents in Germany are approximately at EUR 425 p.m. compared to an average annuity for NRW at EUR 967 p.m (a difference of EUR 542 p.m.). In addition, we also know that Baukindergeld homeowners have, on average, initially higher credit cost than their former rents (a difference of EUR 340 p.m., according to I_11).

On the other hand, both credit and rental costs depend on the type of property acquired (purchasing an existing property is cheaper than building a new one), the size of the living space before and after acquisition as well as the locality of both living situations (with highly populated areas usually having higher property costs and rents). Thus, the hypothesis might certainly be true for some borrowers before weighting in this and other additional evidence.

We argue that the initial prior points to a high portion of medium-income borrowers with lower disposable incomes. Arguing a fortiori for only a slight probability, we assumed that 60 out of 100 medium-income borrowers should benefit from the acquisition in the long run (60:40). For low-income borrowers, we expect the odds to be a lot lower

with most borrowers having similar costs for rentals and property. If we reduce the odds here even further to 20:80 (improbable) and account for the fact that 53.5% of borrowers in the sample are mid-income borrowers (I_1), a prior of 41% looks reasonable to us.

### 3.4.2. Consequent for $h_2$ and $\neg h_2$

We considered the following five categories of evidence for $h_2$ being true and for $h_2$ being false: credit burden (BUR), disposable income (DII), size of the property compared to size of former rental flats (SIZ), price of the property regarding its region (REG) and price of the property regarding its type (TYP). We estimated both consequents for each category and then multiply the odds to derive our final consequents.

We know that the credit burden (BUR) for a comparable sample of Baukindergeld borrowers is higher than the average of the population (I_24). This is somewhat expected on $h_2$, as these borrowers also have a lower income compared to first-time owners in the average population. It is more likely on $\neg h_2$. All other things being equal, we expect that is its at least twice as likely on $\neg h_2$ than $h_2$ (1:2).

We also know that Baukinder households are expected to have higher disposable incomes (DII) in cities after acquiring property, even without the additional grants of the federal program (I_16). This element clearly favors $h_2$, especially since it reports on the facts that $h_2$ presumes to be true. However, it is not necessarily conditioned on $h_2$, since other households, especially with higher incomes, should also benefit from the gap in credit costs compared to rents in cities (none of the households in both samples obtained grants in the loan program). All other things being equal, we expect it is only slightly more likely on $h_2$ than on $\neg h_2$ with odds of 100:90.

Homeowners in general move into larger property (SIZ) when leaving rental living space (I_23) and Baukindergeld household do not differ from the average population in this regard (I_22). Larger living space correlates with higher monthly expenditures and thus a lower disposable income. This is also somewhat expected on $h_2$ since most applicants for these types of loans already have children or apply for loans when they are expecting them. However, it is almost certainly the case if $h_2$ is not true. Arguing a fortiori and all other things being equal, we assumed that the odds are slightly in favor of $\neg h_2$ with 80:100.

The majority of Baukindergeld households acquire property in a region (REG) with similar prices (76%), but they are more likely to move to regions with lower prices than a comparable sample of first-time homeowners with families (19% according to I_18). This is expected on both propositions, but slightly more likely on $h_2$. All things being equal, we think that odds of 100:90 in favor of $h_2$ are a reasonable assumption.

The majority of Baukindergeld households also favor used property (TYP) over new property and used flats of used homes, especially compared to the average sample of first-time homeowners (I_17, I_18). They therefore pay lower prices for the property (I_20, I_21). This is clearly expected on $h_2$ and less likely on $\neg h_2$, since the latter would predict that the borrowers behave as the rest of the population in this regard. We find that odds of 60:40 in favor of $h_2$ are a good conservative estimate for this.

Overall, the following odds were multiplied to calculate both consequents:

$$\frac{P(e|h_2.b)}{P(e|\neg h_2.b)} = \frac{1}{2}\ (BUR)\ \times\ \frac{100}{90}\ (DIS)\ \times\ \frac{80}{100}\ (SIZ)\ \times\ \frac{100}{90}\ (REG)\ \times\ \frac{60}{40}\ (TYP)\ =\ \frac{74}{100}$$

The resulting consequents are 74% for $P(e|h_2.b)$ and 100% for $P(e|\neg h_2.b)$.

### 3.4.3. Posterior for $h_2$

Based on the estimates for both priors and consequents, the following degree of warranted belief in $h_2$ was established:

$$P(h_2|e.b) = \frac{41\% \times 74\%}{[(41\% \times 74\%)+(100\% \times 69\%)]} = 34.0\%$$

The posterior probability for $h_2$ being true is therefore assumed to be at circa 34%, pointing to a rather low credibility of this ToC claim. Out of 100 borrowers, only 34 borrowers are expected to have a higher disposable income one year after moving out of rental space and into their new property.

### 3.5. Warranted Belief in Hypothesis 3

**h₃:** *Home acquisition by medium-income borrowers increases and frees-up rental space in regions with high rents, which stabilizes the rental index in these regions.*

#### 3.5.1. Prior for h₃

The prior for H3 refers to a smaller sample of borrowers whose size is constituted by the postal code of the borrowers when applying for the loan (26% live in areas with high densities and high rents according to I_13). However, $h_3$ not only refers to these borrowers, but to a potential population of households that move into the rental space of the borrowers after they moved out (with re-rented space being cheaper than rents for newly built homes according to I_20). Thus, for $h_3$ to be true over ¬$h_3$, only additional effects matter. That is, only if the borrowers in the sample would not have acquired living space anyway and free-up rental space after moving (88% according to I_32), the hypothesis is true compared to any other explanation. We also know that the rental stabilization effect (oozing effects and moving chains) is more pronounced when these borrowers purchased or constructed new homes compared to used homes (I_31), which we used as our reference class according to the rule of greater knowledge (new homes that would not have been acquired otherwise).

A total of 69% of Baukindergeld households acquiring new homes would have acquired them anyway and 6% did not respond to this question in Weber et al.'s (2022) evaluation study (I_29). By combining the share of Baukindergeld households that acquired a new home earlier or at all (36%) and the share of borrowers that freed up rental space (88%), we estimated that 32 out of 100 borrowers in regions with high rents are expected to contribute to a rent-stabilizing effect. Our prior therefore cannot be higher than 32% (P ($h_3$|b)).

#### 3.5.2. Consequent for h₃ and ¬h₃

We considered the following categories of evidence for and against $h_3$: borrowers moving out of rental space in cities (MOV), the income of borrowers before moving out (INC), price increase in re-rented space (PRI) and borrowers using the loan to acquire used space (USE). All evidence for these categories relates to Baukindergeld households compared to the more general sample of families moving into property.

Regarding the first category of moving chains (MOV), we know that Baukindergeld households are more likely to move into less populated areas than the control groups but that the difference is not large (I_18). This is 100% expected on $h_3$ but not very unlikely on ¬$h_3$ either. We estimated the odds for $h_3$ over ¬$h_3$ at 100:90 in a conservative estimation.

The income of the borrowers (INC) can also be constituted as evidence, knowing that low incomes correlate with lower rents (I_35). We know from the Baukinder sample that these types of borrowers have indeed lower incomes compared to the greater sample of families acquiring new living space (I_34) and are therefore expected, on average, to pay lower rents before moving into the new property. Although this is 100% expected on $h_3$ and less probable on ¬$h_3$, we must account for the fact that this is, at best, an indirect inference as the actual rents before acquisitions are not known. Arguing a fortiori, we cannot provide this evidence higher odds than 100:90 in favor of $h_3$.

Larger price increases for rents in regions that already have high rents (PRI) count as evidence against $h_3$. We know that, in larger cities in NRW, the rents are not only higher, but also show a higher price increase for re-rented space over time (I_33). We therefore expected the same for living space that is no longer occupied by the borrowers. This is

100% expected on ¬h₃, but at least slightly improbable on h₃. Arguing a fortiori, we assumed that odds of 75:100 represent a cautious assumption here.

Although excluded from the prior, we argue that evidence for borrowers moving into used acquired property (USE) should increase the posterior, because the desired effect is expected to be lower but still attested to in the literature (see arguments for and against the estimated prior in the previous section). However, Baukindergeld households are more likely to move into new space over used space. As this is as expected on h₃ as it is on ¬h₃, no effect on the consequent is expected here (odds of 100:100).

Overall, the following odds were multiplied to calculate both *consequent*s:

$$\frac{P(e|h_3.b)}{P(e|\neg h_3.b)} \;=\; \frac{100}{90}\;(MOV)\;\times\;\frac{100}{90}\;(INC)\;\times\;\frac{75}{100}\;(PRI)\;\times\;\frac{100}{100}\;(USE)\;=\;\frac{93}{100}$$

The resulting consequents are 93% for $P(e|h_3.b)$ and 100% for $P(e|\neg h_3.b)$.

### 3.5.3. Posterior for h₃

Based on the estimates for both priors and consequents, the following degree of warranted belief in h₃ was established:

$$(h_3|e.b) \;=\; \frac{32\% \;\times\; 93\%}{[(32\% \;\times\; 93\%) + (100\% \;\times\; 68\%)]} = 30.4\%$$

The posterior probability for h₃ being true is therefore assumed to be at circa 30%, pointing to a rather low credibility of this ToC claim. Out of 100 borrowers in regions with high rents, only 30 borrowers are expected to stabilize rents. By comparison with the entire sample, no more than 8% are expected to cause this desired outcome from the program.

## 4. Discussion

The following section interprets and compares the results of all three causal strands and discusses how the credibility of the claims would improve or deteriorate considering new evidence and primary data as well as changes to the ToC.

### 4.1. Interpretation and Comparison of Posteriors (Research Question 1)

Only one claim out of three passed the burden of warranted belief. While the posterior probability for h₁ is clearly above the credibility threshold of 50% (with 76%), the two other hypotheses failed the test with a posterior of 34% (h₂) and 30% (h₃) (although not drastically to such a degree that makes them entirely improbable). Of these three causal claims, only one claim relates to the entire sample of borrowers (h₂) but failed to provide enough evidence for the majority benefiting from the program.

This means that the implicit assumption of the issuer (homeownership loans contributing to SDG 1 and SDG 11) does not—given what we currently know—apply to most of the borrowers or potential subsequent beneficiaries. On the other hand, none of the causal claims has been de-bunked entirely as, depending on the fraction of the sample addressed, at least 30 out of 100 loans are expected to intervene in the systems in the proposed manner. This is also in line with the original assessment, from which we expected 100% (low-income borrowers) on h₁, 26% (all borrowers) on h₂ and 13% (borrowers in cities) on h₃.

For h₁ (property for vulnerable groups), a high initial probability could be attested to, which only slightly decreased when looking at the evidence (from a prior of 85% to a posterior of 76%). By comparison, both h₂ (higher disposable income) and h₃ (stabilized rents) already started with a rather low prior probability that further decreased when considering the evidence (from 41% to 34% and from 32% to 30%, respectively). As these low priors are a consequence of a lack of data, additional primary data from the borrowers would have the potential to increase these odds.

Using the semantic depiction of probabilities in Section 2.2.3, these posteriors can also be interpreted in plain English as shown in Table 4.

**Table 4.** Depiction of the study results in plain English.

| $h_n$ | P(h\|e.b) | Range in Canon | Depiction in Plain English |
|---|---|---|---|
| 1 | 76% | 60–80% | It is at least *slightly probable* (and maybe even *probable*) that the issuer's homeownership loans lead to property for vulnerable groups from the sample of low-income borrowers. |
| 2 | 31% | 20–40% | It is *improbable* (but maybe only *slightly improbable*) that the issuer's homeownership loans improve the disposable incomes of the borrowers. |
| 3 | 30% | 20–40% | It is *improbable* (but maybe only *slightly improbable*) that the issuer's homeownership loans stabilize rents as a result of the share of borrowers that free-up rental space in regions with high rents. |

All data used and all assumptions made come with uncertainties, of course. Therefore, for Bayesian Reasoning, it is also important to quantify what type of data (see next section) and evidence would tip the scales (all other things being equal). Regarding the latter, $h_1$ would drop below the 50% threshold if new evidence was found that is at least 2.5 times more likely on ¬$h_1$ than it is on $h_1$ (to both overcome the high prior and low consequent on h). For $h_2$ and $h_3$, the situation is reversed. To overcome the bad initial odds, the new and additional evidence must be at least 1.9 times ($h_2$) and 2.3 times ($h_3$) more likely on the proposed hypothesis.

We conclude that, although the selected evidence and our assumptions on their likelihood do matter, minor or even medium changes to the estimated likelihoods for the expected evidence would not have a strong effect on the results. Instead, extraordinary evidence is needed to overcome the odds if no additional primary data from the sample is available that changes the prior odds.

*4.2. Influence of Primary Data on Warranted Beliefs (Research Question 2a)*

The data used in the BA stemmed, for the most part, from the literature rather than primary data by the issuer. Although the evaluation of the federal program of "Baukindergeld" [53] was a useful primary source to that end, only few concrete data points could be established. This uncertainty extends both ways, of course, but it influenced the credibility of the hypotheses, as we argued from a fortiori whenever those uncertainties arose.

For research question 2, we investigated what type of primary data are needed and how this primary data would influence the results of the BA. Table 5 summarizes this investigation. The crucial missing information is mostly centered on the actual income of the borrowers (income distribution) and their monthly expenditures before and after acquisition (especially cold rents in their former rental homes). We expect higher posteriors on $h_1$ and $h_3$ if these data are available. However, this comes at the risk of smaller samples for each of the claims. This does not lead to negative effects on $h_1$ and $h_3$ as both hypotheses already focus on a portion of the borrowers that either benefit or not.

For $h_2$ on the other hand, a negative effect can and is likely to occur even if evidence is found that corroborates the original claim, because a portion of borrowers will have a lower disposable income for several years. Even if the prior can be improved, which is the prior in the study with the lowest data availability, a rebound effect would have to be accounted for that constitutes a negative contribution to the same sustainability goal.

**Table 5.** Required primary data and likely effects on credibility.

| hₙ | Weakly Attested in b | Required Primary Data from Issuer's Sample | Likely Influence on Posterior |
|---|---|---|---|
| 1 | share or number of borrowers belonging to vulnerable groups | distribution of net incomes; first-time ownership; predictors of vulnerability (esp. handicapped, single parents, single income-earners) | increase in posterior as consequents were low due to missing data with the risk of a smaller sample of low-income borrowers |
| 2 | disposable income of all borrowers before and after acquisition | cold rents before acquisition; credit costs after acquisition; housing costs before and after acquisition | ambiguous effect on posterior as portion of the borrowers more likely to be affected positively against a portion of borrowers affected negatively |
| 3 | oozing effect of used home acquisition in cities and rents before acquisition | distribution of net incomes; cold rents before acquisition; fluctuation at the local housing market and building activity at the local housing market | increase in posterior as there was insufficient evidence for lower rents by borrowers and no evidence that used home acquisition contributes to the stabilizing rent effect |

*4.3. Influences of ToC Changes on Warranted Beliefs (Research Question 2b)*

The last research question deals with the conclusions from the BA. Given what we know now, how could the ToC claims be rephrased for a more credible sustainability effect?

Looking at the posteriors alone, only $h_1$ seems to be credible and does not require any major changes. This is partly because the effect of homeownership alone is attested to a priori (only borrowers that applied successfully are in the sample) and is therefore assumed to be an additional effect. A useful expansion of the claim could be to include borrowers that would not have been able to acquire a home otherwise. Such a group of borrowers would improve the credibility even further, even if this means that the portion of beneficiaries decreases consequently (higher posterior for a smaller sample).

For $h_3$, we found a low credibility of the ToC claim against evidence from the literature [63,64] that the desired outcome in fact occurs if certain conditions are met. This discrepancy could be, at least to a degree, resolved if the claim is re-defined considering this evidence. Subsequent households that move into the former rental space of new homeowners benefit in terms of rents if these acquired homes can be considered an upgrade compared to the former living condition (ibid). This is particularly the case for newly built properties, but it not restricted to this (ibid). The ToC claim should consider this mechanic and restrict its effect to the sample of borrowers that move into homes with higher relative (per m²) and absolute costs of property.

For $h_2$, even a higher credibility of the claim would have to be argued against cases in which the disposable income is lower after acquiring property. Instead of being re-defined, it should be either (a) dropped from the ToC entirely, (b) integrated into the entire sample as a portion of borrowers or (c) replaced by a claim that focuses on the accumulation of capital rather than expenditures. Given the available data and what we now know for the other hypotheses, option (b) seems to be the most promising alternative. If we assume that the sample of borrowers for both $h_1$ and $h_3$ could be better attested to, we can also assume that a large portion of the remaining borrowers do not belong to a vulnerable group and do not upgrade their living situation. This portion of the sample therefore relates to borrowers with higher incomes, who avoid higher living costs. We expect that a BA for this fraction of the borrowers would result in a higher posterior.

We also provided some insights into the loan program itself. As already stated by some of the authors cited in the study, the advantages of successfully obtaining a loan in the first place as well as lower financial transaction costs are not the most important

drivers of more affordable housing. We know, for example, that most borrowers would have acquired the property anyway. We also know that these borrowers had higher relative capital costs compared to their income than other families, have higher shares of borrowed capital and needed longer to save income for the required equity. A more successful loan program from a promotional bank could intervene here by taking over financial risks of households or providing the equity itself in forms of grants for low-income families.

### *4.4. Limitations*

The BA revealed several limitations of the ToC model. One weakness of the ToC approach shown here is its simplicity regarding the systemic changes it describes. A more complex and non-linear logic model would include additional emergent effects, an explication of actors and the definition of external pre-conditions for change. This would certainly result in more credible claims, but it would also be more time-consuming and would require expertise in each area it investigates. Such complex models would defy the purpose of impact reporting. Impact reports for bonds usually cover several social dimensions and are intended to be quick (published no later than one year after the emission). It can, and should be, considered for a more scientific assessment, especially if several impact reports concerned with the same issue are to be evaluated.

In opposition, the BA also showed how fragile the claims can become if even simpler causations are assumed. A sole SDG mapping, for example (as it is usually suggested in frameworks pre-dating the impact reporting), would not include propositions on the direct activities and results of the main actors involved. The same is true for the direct use of indicators from input-/output models or the corroboration by SPO providers. Since almost no explanation is given in these cases, a BA would result in low priors (large samples with non-relevant classes) as well as equal probabilities for both consequents.

We think that the main limitations of a BA were well managed in this study by employing the rules of greater knowledge and from the stronger argument. It is of course possible to come to different conclusions using different evidence and probability estimates. As Glymour (2016) [65] stated in his arguments against the use of Bayesian confirmation theories, "Whereas the ideal Bayesian agent is a perfect logician, none of us are, and there are always consequences of our hypotheses that we do not know to be consequences" (p. 149). Such an alternative would not be more reliable by default though, as its results would also only inform on the degree of warranted belief, given what we currently know. Even if such a BA would come to an entirely different conclusion (which we do not expect), it would not speak against the use of the method per se. Knowing more or understanding better the circumstance under which such a loan program lead to desired results, is always better than having no indication at all. Additionally, since each BA always provides information on the probability of alternative explanations, there is no absolute certainty anyway.

## 5. Conclusions

This study investigated whether causal claims in the market for Social Bonds are credible. The case study used was an implicit stakeholder theory for homeownership loans that was explicated with a Theory-of-Change and tested with the help of a Bayesian Analysis. Both methods combined provided indeed evidence, or at least a warranted belief, that one causal strand out of three is credible. However, it also questioned whether the remaining two impact pathways play out as the ToC claims they would. From the analysis shown here, it is almost certain that the fraction of beneficiaries is considerably lower than proposed from the narrative alone. It is also probable that there are also borrowers that will experience negative effects such as a lower disposable income. By comparison with the original indicator quantification, the estimated posteriors are well within the margins of error. There is only a small risk of social washing from the impact reporting, because its reported effects are already limited to fractions of borrowers.

The results of the original impact report, for which the ToC was developed, now come with higher uncertainties. Nonetheless, we consider this to be an improvement, especially because the combination of both methods pinpointed to the weaknesses in the original argument and its data. This allowed us in turn to identify not only what type of data would improve the odds of warranted belief, but also how the causal claims could be rephrased to represent a better picture of reality (and the desired outcomes of the issuer).

Another crucial weakness of ToCs for impact reporting are biases by the analyst. One example for that are pattern-seeking biases involved in the process. By accepting the claims of the issuer (specific inputs contributing to explicit societal goals), the analyst actively looks for causations that might explain that claim. Any other plausible causational mechanic is usually neglected, especially if it might lead to negative effects. It is difficult to imagine how this bias can be entirely avoided, given that the analyst must not be involved in the underlying framework of the bond (selection of eligible assets and definition of goals). The BA helps to mitigate this issue though. The formulation of hypotheses to be tested, and definition of reference classes for priors and especially the consideration of consequents if any other explanation is true, enables the analyst to spot at least the most obvious fallacies in his argument. Since BA requires a more stringent logical argumentation than a ToC alone, it is also reasonable to assume that other forms of biases, such as cognitive or confirmation biases, are mitigated as well.

However, the sheer amount of work and data needed to conduct a BA properly in conjunction with a ToC is indeed a limitation that needs to be discussed. The case study here dealt with only one program in one out of four areas of intervention (with the subsequent bond including more projects and areas of intervention). A full BA for each of these areas can therefore not be considered a feasible solution to the question of credibility of Social Bonds. We hope we can show in the future how the approach can be simplified without foreclosing on its advantages. A promising starting point for that could be flowcharts for Bayesian Reasoning without math as suggested by Carrier (2012) [46]. Such an approach could build on the fact that the challenge of attributing effects to interventions and changes in the system is easier for some parts of a ToC than others. The causal connection between outcomes and impacts, for example, is usually easier to ascertain, or at least to argue, than outputs on project level causing persistent changes for entire social groups. Showing how these different levels of plausibility interact in Bayesian terms could provide analysts, issuers and investors with a quick understanding of the pitfalls of impact reports. However, since such a solution would rely on less empirical data, it would have to be grounded more firmly on theoretical grounds for causation. The ongoing work by Judea Pearl, Clark Glymour, Richard Scheines and others on causal models could provide such a basis [66–69]. Although the causal hypothesis of ToC, such as that shown in this study, does not contain explicit assumptions on confounding or mediating effects, the process of investigating whether such effects exist and how they influence the robustness could be captured by control questions and decision trees.

**Author Contributions:** conceptualization, J.T.; methodology, J.T.; validation, S.S.; formal analysis, J.T.; investigation, J.T. and S.S.; data curation, J.T.; writing—original draft preparation, J.T.; writing—review and editing, J.T. and S.S.; visualization, J.T. All authors have read and agreed to the published version of the manuscript.

**Funding:** We acknowledge the financial support by Wuppertal Institut für Klima, Umwelt, Energie gGmbH within the funding program Open Access Publishing. While this research received no external funding, the original impact assessment (that was reviewed here) of the NRW.BANK Social Bond (#2) was commissioned by NRW.BANK.

**Institutional Review Board Statement:** Not applicable.

**Informed Consent Statement:** Not applicable.

**Data Availability Statement:** The underlying impact method can be found at https://wupper-inst.org/en/p/wi/p/s/pd/1885.

**Acknowledgments:** We thank NRW.BANK for its data, consent and support to improve our impact methods.

**Conflicts of Interest:** The authors declare no conflicts of interest. The funders had no role in the design of the study; in the collection, analyses, or interpretation of data; in the writing of the manuscript, or in the decision to publish the results.

## Appendix A

*Appendix A.1. Entities of the Theory-of-Change (ToC) for NRW.BANK Social Bonds*

This operationalization here (as well as methods for investigating causality claims) are further discussed and developed in an on-going dissertation by the corresponding author (Teubler, J.: "Logic Model for ESG Impact Pathways and Assessments".

In the ToC framework for NRW.BANK Social Bonds, inputs are the interventions by the bank in the change system and impacts the sustainability goals the interventions work towards. Activities refer to task performed by the target groups (normally some form of materialization) and outputs are the direct results from these actions. Inputs, activities and outputs describe the project level, where the external actor (here the bank) can directly or indirectly influence the results (described by a responsibility ceiling in the final ToC). Beyond that point, the context changes and outcomes on the societal level are aimed at. This require additional efforts by other actors, but these efforts are not explicitly mapped (but considered for the narrative, hazards and rebounds). Intermediate outcomes improve the lives of beneficiaries (regions or groups) in one way or another, but only long-term outcomes describe persistent changes that are needed for a contribution to impacts.

The ToC is drawn in six steps from the outside to the inside: (1) defining the impacts and long-term outcomes (here sub-targets of SDGs), (2) describing the actors on intervention side of the pathway (here the bank and the target groups), (3) defining the tasks performed by these target groups, (4) investigating desired intermediate outcomes as preconditions for change and possible outputs of activities, (5) map plausible outputs as effects of activities and causes for desired outcomes, and (6) writing a plausible narrative for change.

The final assessment then consists of four additional steps: (7) identifying potential rebounds and hazards, (8) identifying potential indicators of change (thought of as effects positioned on the arrows between the components), (9) measuring or estimating quantifiable metrics with the help of data, models and assumptions, and (10) assigning an indicator quality to all quantified effects based on their position in the outcome pathway. The latter is a grade-scheme ranging from A for measuring long-term outcomes (best-needed indicators) to E (minimum reporting standard) for merely quantifying the inputs (e.g., amount of money for certain target groups).

*Appendix A.2. BA components in the context of the study*

The BA relies on the *hypothesis h*, the *background knowledge b* and the *evidence e*. Because the prior probabilities of all hypotheses sum up to 1 (P(h|b)+P(¬h|b)=1), three probabilities must be estimated:

1.  The *prior probability* (between 0 and 1) of h being true based on b and compared to a suitable *reference class*.
2.  The *consequent probability* (between 0 and 1) of e being present if h is true.
3.  The *consequent probability* (between 0 and 1) of e being present if h is false.

The background knowledge b constitutes everything we (currently) know about the conditions in which h is true in general (h|b) and everything we learned specifically that would predict evidence e to be present if h or ¬h is true (e|h.b and e|¬h.b). These two datasets are exclusive (evidence cannot be used for priors and probability of a general case is not evidence), but the final selection of which goes where is not. A certain information

(e.g., the results of a study) can be removed from e|h.b and transferred to h|b for example without affecting the (final) posterior probability, if this change is accounted for in both priors and consequents. It is also possible to calculate the results from a sequence of BAs, where the posterior of BA1 is used as prior in BA2.

The prior probability or prior is a measure for the number of times the hypothesis is true compared to the number of times it is not, which is why it can also be described by odds. Odds are expressed as fractions that are normalized for their probability via the full sample. If event A is x times more likely than Event B (y), the probability P for odds of x:y is calculated as x/(y+x). The prior can be compared to the chance that a certain event occurs given what we know about similar events. This depends not only on our knowledge of such events, but also on our definition what characterizes a similar event. The latter is called the *reference class* and can also be understood as the context in which the investigation takes place. Using an example from Carrier (2012), the hypothesis "Most royal flushes in a poker game are fairly drawn" does not refer to the overall chance of a royal flush being drawn (which is very low). Rather, it refers to cases where a royal flush is drawn by fair chance (which is rather high) compared to the total sample where royal flush are drawn by other means as well (e.g., slide-of-hand).

It is often possible to use different reference classes. As this decision can strongly affect the results of a BA, two conventions apply when selecting a reference class (see also "Determining a Reference Class" in Carrier (2012, p. 229 et sqq.) for a more thorough investigation of the issue): (A) rule of greater knowledge and (B) rule from the stronger reason (argumentum a fortiori, or just a fortiori). Rule (A) states that, between two potential reference classes, the one that is more relevant for the hypothesis must be selected. In the study at hand, a specific statistical result for the State of NRW (where the intervention takes place) constitutes greater knowledge than the same statistical result for Germany or the entire world. The same is true for results from the federal loan program "Baukindergeld" compared to more general data on homeownership. Although Baukindergeld loans include additional grants to the borrower and are restricted to families with children, they use the same income ranges for applicability as the NRW.BANK loans (which include borrowers that are singles).

If the rule of greater knowledge (A) does not apply, because information is missing or it is unclear which reference class is to be preferred, rule (B) is used instead. Rule (B) states that if in doubt, and all things being equal, the reference class should be selected that results in the lower estimate of the prior probability for h (hence the more conservative estimate for the hypothesis to be tested).

The two consequential probabilities or *consequents* are independent of another in the sense, that they can both be low or high (between 0 and 1). Therefore, if *e* is as likely to be present for both h1 and ¬h1, it has no effect on the result of the BA. Only if P(e|h.b) >> P(e|¬h.b), low prior probabilities can be overcome. This is often translated into the truism "extraordinary claims require extraordinary evidence" (Carl Sagan). However, BT also applies to the logical deduction of this statement as "ordinary claims require only ordinary evidence" (attributed to Gwern Barnwen [70])). Consequents are estimated in a very similar way to the priors. Their probability space (reference class) refers to all logical possible consequences that can be predicted by h and are mutually exclusive of each other [46]. The important distinction to priors is the question a consequent responds to compared to the prior. Instead of asking how likely it is that the proposition is true, it asks how expected the evidence is, if it is true (P(e|h.b)) and how expected it is, if something else caused it (P(e|¬h.b)). In addition, consequents can consider evidence that is conspicuously not there (which would lower the consequent). It is also possible and sometimes worth considering the inverse of the consequents as well. The answer to the question "How often would the hypothesized facts not produce e?" directly relates to P(e|h.b) since P(e|h.b)+P(¬e|h.b)=1 [46].

*Appendix A.3. Background knowledge used in the study*

**Table A1.** Background knowledge used in the study.

| Element ($I_n$) | Use in Study |
|---|---|
| I_1    A total of 46.5% of the loans go to borrowers with a taxable household income below TEUR 50 (gross income, defined as "low-income" by the issuer that also provided the data). | $h_1.b$; $h_2.b$ |
| I_2    The ratio between taxable incomes in NRW compared to net equivalent incomes was 44.3:100 in 2015 on average [51], which means that the low-income threshold of the issuer can be compared to a equivalent income of EUR 22,150 per person and year or EUR 1,845 per month for a single (factor 1), EUR 1,419 p.m. for one parent with one child (factor 1.3), EUR 1230 p.m. for a couple (factor 1.5) and not more than EUR 1,025 p.m. for couples with one child and more (factor 1.8). | $h_1.b$ |
| I_3    The threshold for at-risk-of-poverty in NRW (2020) is at an equivalent income of EUR 1123 p.m. for a single-household, EUR 1460 p.m. for single parents with one child, EUR 1684 p.m. for couples without children and EUR 2021 p.a. for a couple with one child [61]. Only singles in the low-income sample of the issuer are therefore (see h1.b1, h1.b2) not below that threshold. Any other family type can be defined as "vulnerable group" (notwithstanding the fact that other borrowers could belong to that group as well). | $h_1.b$ |
| I_4    The share of first-time homeownership in Germany (2016–2017, without property from inheritance or donation) by families with children is at 54% compared to 13% for singles and 33% of couples without children [55]. | $h_1.b$ |
| I_5    A total of 71% of first-time owners (Germany, 2018) lived in rented living space before acquiring property [71]. | $h_1.b$ |
| I_6    A total of 23.5% of tenants in Germany (2020) spent more than 40% of their net income for housing costs compared to 19.8% of home owners with mortgages/loans and 12.3% of home owners without these financial obligations [72]. | $h_2.b$ |
| I_7    The average share of housing costs in larger cities in Germany (2018) in relation to net income is at 26% for tenants and 21% for home owners [73]. | $h_2.b$ |
| I_8    The owner occupancy costs for homeowners are 38% lower than for tenants in 2020 in Germany (4.3 EUR/m² compared to 7.0 EUR/m² according to [59]). The occupancy costs are defined by the authors as cold rent (tenants) compared to costs for acquisition and maintenance (homeowners). However, the actual repayment portion of home owners are, in opposition to interest costs, not included in this metric. We therefore deem owner occupancy costs only to be relevant for the hypothesis in the long run. | $h_2.b$ |
| I_9    The cold rent (gross) in Germany (2018) ranges from 335 EUR/month (net household income of less than 900 EUR/month) to 1084 EUR/month (income of more than 6000 EUR/month) [60]. The average cold rent (gross) for NRW was at 525 EUR/month [51]. | $h_2.b$ |
| I_10   The annuity (annual costs for interest and repayment for 100 square meters with 25 years loan term, full repayment loan, 20 percent equity) of property in NRW (2021) ranges from 5692 EUR/a (1 county) to 15,428 EUR/a (3 counties) [59], which translates to monthly expenditures of circa 474 to 1286 EUR/month. Overall, annuity in Germany is at 11,609 EUR/a or 967 EUR/month (ibid). | $h_2.b$ |
| I_11   The monthly credit costs (repayment and interest) of participants in the federal program "Baukindergeld" were, on average, EUR 340 higher than their rental costs before moving compared to EUR 650 for other families [53]. | $h_2.b$ |
| I_12   Almost all Baukindergeld borrowers were tenants before home acquisition (at least 98% according to [53]). | $h_1.b$; $h_2.b$ |
| I_13   One quarter of the loans (26%) went to borrowers living in high-density areas with more than 1500 citizens per square-km (postal codes of borrower provided by issuer; allocation as part of the assessment in [33]). | $h_3.b$ |
| I_14   Rents for re-rented space (standing rents) are circa 25% lower than first-time rents or rents for new buildings in NRW (2018/2019; more than 10.2 EUR/m² for new rentals compared to an average of 7.5 EUR/m² for re-rental according to [56]). | $h_3.e$ |

| Element ($I_n$) | Use in Study |
|---|---|
| I_15 "Baukindergeld" homeowners save disposable income compared to renting flats in the same city in Germany (2018) according to [54]. This effect is stronger for low-income households and persists (ibid) even without the additional grants per child provided by Baukindergeld, but not NRW.BANK loans. | $h_2$.e |
| I_16 The comparison with families who moved into property between 2015 and 2018 shows that Baukindergeld households form homeownership disproportionately in less expensive regions and are underrepresented in regions with high purchase prices [53]. | $h_2$.e |
| I_17 A total of 95% of Baukindergeld households either stays in the same area of density (76%) or moves to a less populated area (19%). These are higher rates than comparable families with 79% staying in the same type of region and only 16% moving to less populated areas [53]. | $h_2$.e |
| I_18 A total of 11% of Baukindergeld households acquired a used flat, 54% a used house, 3% a new flat and 32% a new house [53]. By comparison with the larger sample of families moving into acquired property, they are less likely to move into used space. | $h_2$.e; $h_3$.e |
| I_19 The price for used flats in NRW (2019) is circa 52% lower than for new flats (1,720 EUR/m$^2$ compared to 3,260 EUR/m$^2$ according to [56]). For houses the price is circa 16% lower (EUR 319,000 compared to EUR 378,500). | $h_2$.e; $h_3$.b |
| I_20 The overall costs of used property acquired by Baukindergeld households is 18% lower than the total costs of new property (EUR 329,000 compared to EUR 401,000 according to [53]). | $h_2$.e |
| I_21 Baukindergeld households do not occupy a larger property than the average population [53]. | $h_2$.e |
| I_22 Households in Germany (2013–2017) that move out of rental flats into acquired homes increase their living space by 22 m$^2$ on average [58]. | $h_2$.e |
| I_23 The credit burden of Baukindergeld households is higher than the credit burden of a comparable sample of first-time homeowners in Germany (27–32% compared to 20–25% depending on the region and according to [53]). | $h_2$.e |
| I_24 The age of Baukindergeld households is slightly lower compared to first-time homeowners in Germany with children (36.4 compared to 38 according to [53]). | $h_1$.e |
| I_25 A total of 12% of Baukindergeld households are not German nationals, compared to a share of 15% for families in German [53]. | $h_1$.e |
| I_26 The shares of Baukindergeld households with 1, 2, 3 or more children do not differ greatly from the overall sample of families that moved into their own property [53]. | $h_1$.e |
| I_27 A total of 23% of Baukindergeld households have a taxable income of less than TEUR 30 and 76% of less than TEUR 60 [53]. | $h_1$.b |
| I_28 A total of 69% of Baukindergeld households that moved into new homes would have acquired property anyway, while 6% did not respond to this question according to [53]. | $h_3$.b |
| I_29 A total of 59% of Baukindergeld households that moved into used property would have acquired property anyway, while 9% did not respond to this question according to [53]. | $h_3$.e |
| I_30 Acquisition of property by families usually frees up more affordable living space for others (Braun 2020 in [53]). This desired effect can be curbed by price increase in these homes and is most effective if families move into new homes (35% of the Baukinder households) rather than used houses and apartments (ibid) and if the borrowers move into areas with another price range (e.g., from cities into urban areas). | $h_3$.b |
| I_31 A total of 8% of Baukinder households have previously rented the space they acquired later and do not contribute to freeing up more affordable living space. Overall, rental space was freed up in 88% of all cases (Braun 2020 in [53]). | $h_3$.b |
| I_32 The rental costs for re-rented space increased in NRW in general between 2009 and 2019 [56]. This rate is higher for already highly priced cities such as Cologne, Düsseldorf and Aachen compared to lower priced cities and regions such as Dortmund, Duisburg or the county of Höxter (ibid). | $h_3$.e |
| I_33 Baukinder households have, on average, lower incomes than the larger sample of families moving into acquired property [53]. | $h_1$.b |

| Element (I_n) | Use in Study |
|---|---|
| I_34   Baukinder households are more likely to save equity capital before acquisition (61%) compared to the average population of first-time owners (37%). However, the average savings period is the same [53]. | h_1.e |
| I_35   The households in NRW (2018) with the lower third of incomes pay less rent but pay a higher share of their income for theses rents [51]. | h_3.e |

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
