# Peer review of "Causal Strands for Social Bonds—A Case Study on the Credibility of Claims from Impact Reporting"

_sustainability, doi:10.3390/su141912633_

Round 1
Reviewer 1 Report
The research issue is novel and the findings contribute to the literature. I suggest authors integrate theoretical perspectives and critically compare study results with past findings to highlight the paper's originality.
Author Response
Dear Reviewer,
we thank you kindly for the review of our manuscript. We hope that our revised version is now in line with your expectations.
As the suggestions and criticism by all reviewers were very similar, we decided to write a single response on the following identified issues identified in your review(s):
- Insufficient integration of theoretical perspectives, high ratio of non-peer-reviewed literature and lack of comparison to past findings
- Lack of literature review and conclusions
- Over- and/or misuse of footnotes, quotation marks and italic emphasis
- Overall problems with editing, grammar and spell-checks
ad 1)
As described in the introduction, theoretical background on impact reporting for sustainable-labelled bonds is non-existent. One reason for that it is highly commercialized and requires transdisciplinary applied-science approaches. Nobody has, to our knowledge, used Theories-of-Change to identify impact indicators for bonds before or applied Bayesian Reasoning to test the claims of bond frameworks and SPOs. This is, of course, also the reason why we started with a method paper in the first place. If ToCs are not relevant or applicable to the question of impacts from bonds, we like to know that before advancing the methods further.
Although some core-concepts have been discussed in literature, this literature consists, for the most part, of working papers/reports and focuses on impact investing. Another important issue is the fact that the original theoretical framework for Theories-of-Change has diversified into scholarly theories concerned with sociological impact-pathways (with a strong focus on monitoring human capacities and competences) of developing projects versus numerous practitioner solutions more concerned with organisational change and storytelling. These, more applicable solutions, are under investigated in professional literature as well.
However, we agree that we did a poor job in summarizing the literature and put additional efforts into finding peer-reviewed literature that is at least related to the questions raised. In cases where no peer-reviewed literature was available, we tried to cite the most relevant grey literature.
Other than that, literature used in the main part of the study mainly consist of statistics and data from evaluation reports on a very similar loan programme. We think that using this kind of data is appropriate, as the data used relates more closely to the interventions than broader studies.
ad 2)
We introduced the following sections to the introduction (that is now sectioned into 1.1 literature review and 1.2 research questions):
- Studies in the area of Green Bonds/Sustainable Finance with a focus on environmental impacts, and the effects/correlations of/with certification and reporting
- Indicators and impact assessment methods for Green or Social Bonds
- Application of Theories-of-Change for impact investing
We also distinguished more clearly between the discussion of the results, limitations and research questions compared to our overall conclusions on the value of the method(s).
Discussion 4.1 to 4.3 now explicitly address the 3 questions and 4.4 the limitations of the study and methods used.
The conclusion (5) now summarizes the results, provides insights into the secondary findings from using a case-study on homeownership loans and interprets the original claims of the ToC in light of these findings. It further addresses the question of biases by the analysts and how these biases were at least mitigated in our opinion. We then conclude with an outlook on our future research, given that the approach here is a solution, but not a very pragmatic one. Drawing on literature used in the study, as well as the lead authors research on the use of diagrams for causal inference (mostly based on J.Pearl and others), we postulate a decision-tree approach that accommodates implicit confounding or mediating factors in the ToCs.
ad 3)
We either integrated or removed almost all footnotes in the manuscript. We have reduced the number of footnotes to 8, of which only 2 appear in the main manuscript.
We also removed almost all italic emphasis as well as bold captions from the manuscript. The use of italics is now restricted to the appendix section on Bayes Theorem and used to introduce the terminology. As these tools were used for different purposes and are therefore not obvious to the reader, we agree that it is better to remove them as much as we could.
ad 4)
Although the journal prefers numbered references, we decided to use a Harvard style for our first submission, as we find that this makes it easier to review (and revise) the text. The citations have now been converted in numbered references; and combined with Harvard-Citations only where it is necessary (mostly in the literature review).
Since the original manuscript was already very long, we tried to compensate our new additions with a more concise style as much as possible. We also worked on the readability of the manuscript and deconstructed some longer sentence constructions in the original draft.
All texts should now also comply with American English.
kind regards,
Jens Teubler & Sebastian Schuster, September 18th 2022
Reviewer 2 Report
‘Although Social Bonds are’, ‘Social and Sustainability Bonds’, etc. – why initial capitalization? "future contract on social outcomes" (Stellina Galitopoulou & Antonella Noya, 2016, 54 p. 4), ‘OECD defines blended-finance as "the strategic use of development finance for the mobilisation of additional finance towards sustainable development in developing countries" (OECD, 2018, p. 16), p.16’, etc. – are italics in the original or added by you for emphasis? Clarify this. The manuscript must be inspected by an editorial service. Try and provide more references to support your ideas that are typically substantiated by only one source – and as recent as possible. ‘the increase in “green” capital’ - Say directly the words you mean, not in quotes, as their meaning is thus unclear (sometimes the opposite one is got). Footnotes should be integrated in the text. ‘hypothesizes’ – ‘hypotheses’. Research questions and hypotheses must be constructed based on specific supporting sources, preferably as recent as possible. ‘1990ies’ – ‘1990s’. You should compare your results with others in terms of concrete data for better research integrative value. The manuscript requires major revisions to contextualize the merits of the study and potential uses of its methodology in future studies. ‘as suggested by Richard Carrier in "Proving History"’ – remove. The missing Conclusion section should clarify the main contribution of the paper and the value added to the field. Please provide more details regarding the study limitations and strengths and what this means for the study findings.
The proportion on non-peer reviewed sources is unacceptably high. Here are some recent suggestions:
Kliestik, T., Valaskova, K., Lăzăroiu, G., Kovacova, M., and Vrbka, J. (2020). “Remaining Financially Healthy and Competitive: The Role of Financial Predictors,” Journal of Competitiveness 12(1): 74–92. doi: 10.7441/joc.2020.01.05.
Priem, R. (2021). “An Exploratory Study on the Impact of the COVID-19 Confinement on the Financial Behavior of Individual Investors,” Economics, Management, and Financial Markets 16(3): 9–40. doi: 10.22381/emfm16320211.
Lăzăroiu, G., Ionescu, L., Andronie, M., and Dijmărescu, I. (2020). “Sustainability Management and Performance in the Urban Corporate Economy: A Systematic Literature Review,” Sustainability 12(18): 7705. doi: 10.3390/su12187705.
Author Response

(The authors gave the same response as above.)

Reviewer 3 Report
My comments:
1. The topic of this paper is interesting and it will contribute in related research field.
2. A section of “Related Works” or “Literature Review” is necessary for this paper.
3. The title of the figures should be indicated below the figures.
4. I can't find the conclusion, what’s wrong? A section of “Conclusion” is very important for an academic paper. For example, the contributions to academic research as well as theoretical implications, research limitations, and suggestions for further research.
Author Response

(The authors gave the same response as above.)

Round 2
Reviewer 2 Report
This revised version can be published.
Reviewer 3 Report
This paper is qualified to be published.